# Fish Community Structure and Biomass Particle-Size Spectrum in the Upper Reaches of the Jinsha River (China)

**DOI:** 10.3390/ani12233412

**Published:** 2022-12-04

**Authors:** Taiming Yan, Jiayang He, Deying Yang, Zhijun Ma, Hongjun Chen, Qian Zhang, Faqiang Deng, Lijuan Ye, Yong Pu, Mingwang Zhang, Song Yang, Shiyong Yang, Ziting Tang, Zhi He

**Affiliations:** 1College of Animal Science and Technology, Sichuan Agricultural University, Chengdu 611130, China; 2College of Environmental Science and Engineering, Guilin University of Technology, Guilin 541004, China

**Keywords:** freshwater ecosystems, fish biodiversity, biomass, Jinsha River basin

## Abstract

**Simple Summary:**

From 2019 to 2020, 28 species of fish were collected from the upper reaches of the Jinsha River. The dominant species of fish were *Triplophysa stenura*, *Schizothorax wangchiachii*, and *Schizopygopsis malacanthus*. Fish diversity was at a deficient level, and flow velocity, altitude, and dissolved oxygen were the main influencing factors causing the differences in fish community structure in the upper Jinsha River. The abundance/biomass curve showed that the fish community in the upper Jinsha River was moderately or seriously disturbed. The standardized biomass size spectrum of fish showed that the degree of disturbance of fish in tributaries was much lower than that in the main stream. Compared with the historical data, the structure of the fish community in the Jinsha River changed significantly, with more exotic species and smaller individual fish. It is suggested that habitat conservation strategies be adopted in the upper tributaries of the Jinsha River to provide a reference for the restoration of fishery resources and the conservation of fish diversity in the Yangtze River.

**Abstract:**

To understand the characteristics of the fish community structure and biomass particle-size spectrum in the upper reaches of the Jinsha River, fish and environmental surveys were conducted in 21 segments of the upper reaches of the Jinsha River in September 2019 and June 2020. A total of 4062 fish belonging to 2 orders, 5 families, 18 genera, and 28 species were collected. Among them, Cyprinidae fish were the most abundant (14 species), accounting for 50.00%. The Shannon index and Pielou evenness index values varied from 0.402–1.770 and 0.254–0.680, respectively. The dominant species of fish were *Triplophysa stenura*, *Schizothorax wangchiachii*, and *Schizopygopsis malacanthus*. Redundancy analysis (RDA) was used to analyse the relationship between the fish community and environmental factors. Velocity, altitude, and dissolved oxygen were the main influencing factors of fish community structure differences in the upper reaches of the Jinsha River. The abundance/biomass curves showed that the fish communities in the upper reaches of the Jinsha River were moderately or severely disturbed. The standardized biomass particle-size spectrum of fish showed that the degree of disturbance of fish in tributaries was much lower than that in the main stream. Compared with the historical data, the fish community structure in the Jinsha River has changed significantly, with the number of exotic species increasing, and the individual fish showing miniaturization and younger ages. It is suggested that habitat conservation strategies be adopted in the upper tributaries of the Jinsha River to provide a reference for the restoration of fishery resources and the conservation of fish diversity in the Yangtze River.

## 1. Introduction

The upper reaches of the Jinsha River are located in the Hengduan Mountain Region of China; this area has a high degree of biodiversity and is considered 1 of 34 biodiversity hotspots worldwide [1]. The upper reaches of the Jinsha River provide important support for the development of the Yangtze River economic belt. In freshwater ecosystems, fish are one of the most sensitive and reliable indicators of ecosystem function, and maintaining fish diversity is the basis of sustainable fisheries development [2]. However, with rapid economic development and hydropower development, the environmental pollution of water bodies is aggravated, leading to habitat fragmentation, spatiotemporal changes in runoff distribution, increased disturbance of the natural habitat environment of fish, and serious threats to river ecosystems. In addition to overfishing, fish resources and diversity in the upper reaches of the Jinsha River have shown a declining trend [3,4,5].

At present, the study of fish species in the upper reaches of the Jinsha River is limited to the composition of fish species, and there is no report on the relationship between fish community changes and environmental factors. First-level fish fauna were studied in the Jinsha River basin as an independent study area by Wu and Wu [6], who reported that the fish fauna composition in the Jinsha River basin was simple and mainly composed of Schizothoracinae and Sisoridae. Since then, ‘The Fishes of the Hengduan Mountain Region’ [7] and ‘Fuana Sinica Osteichthyes Cypriniformes II’ [8] projects have been carried out to conduct a comprehensive analysis of the fish fauna in the region. In a survey of fish resources in western Sichuan, Yang et al. summarized the distribution characteristics of plateau fish in the Shiqu and Ganzi regions of the Jinsha River trunk stream [9]. Hu conducted a survey of fish stocks in the upper reaches of the Jinsha River in 2010–2011 [10]. The latest systematic survey and research report on the upper Jinsha River came from ‘Fishes in the Jinsha Jiang River Basin, the upper reaches of the Yangtze River, China’, and the survey results showed that 16 species of fish were collected in the upper Jinsha River [11].

In aquatic ecosystems, the fish-size spectrum has been used to assess the production status of different trophic levels and to predict the impacts of various human activities on the ecosystem [12,13]. The particle-size spectrum concept was first proposed by Sheldon and Parson [12], and this approach represents the curve of the relationship between biomass or the number of organisms (i.e., abundance) [13] and the size of a species. It also reflects the characteristics of biomass, abundance, and size structure through curve fluctuations, making it an important method for aquatic ecosystem research [14,15,16]. Since particle-size spectrum theory was introduced into aquatic ecosystem research, the fish particle-size spectrum has been widely applied to evaluate how different trophic level layers affect fish biological community structure characteristics and the impact of human disturbance on the ecological system [13]. For example, Duplisea and Kerr [17] used fish particle-size spectrum parameters to illustrate the effect of natural disturbance on deep fish community structure. Graham et al. [18] studied the impact of fishing on coral reef fish communities through the fish-size spectrum and found that fish abundance was negatively correlated with fishing intensity within the range of grain size. Blanchard et al. [19] used four fish populations in the North Sea as examples to establish a particle-size spectrum model to assess the response of the community to fishing and to monitor the environment. Studies on the particle-size spectrum of plankton [20,21] and marine benthic organisms [22,23] have been extensive and occurred relatively early, but studies on the particle-size spectrum of fish were reported successively in recent years [13,24].

The majority of fish distributed in the upper reaches of the Jinsha River depend on the flowing water habitat for the whole or part of their life history (such as reproduction) [6,11]. The construction of cascade power stations in the upper reaches of the Jinsha River will lead to habitat barriers and fragmentation, shrinkage of flowing water habitats, and changes in the natural hydrological situation [10]. The tail of the main stream reservoir, the local waters of tributaries in the reservoir area, and even the mouth of tributaries will become important areas for fish to reproduce and inhabit [3,4]. Based on this, this research used survey data from 2019 to 2020 for fish and the environment, analysed the fish diversity status upstream of the Jinsha River, and built a fish biomass particle-size spectrum for the main stream and tributaries. The study also discussed the relationship between community and environmental factors and clarified the habitat effect of fish tributaries under cascade development in order to provide reference for fishery resource restoration and fish diversity conservation in the upper reaches of the Jinsha River.

## 2. Materials and Methods

### 2.1. Study Area

The upper reaches of the Jinsha River range from Zhimenda, Qinghai, to Shigu, Yunnan, covering an area of 76,500 km^2^. The reach is 974 km long, with an elevation drop of 1715 m and an average specific drop of 1.76‰. The temperature is low in the north and high in the south. The annual average temperature is −4.9~7 °C in the north and 12~15 °C in the south. The area of the watershed is 5.1 × 10^4^ km^2^, of which grassland occupies the largest area (45.18%), woodland occupies 28.97%, shrub land occupies 17.5%, unused land occupies 5.24%, water and water conservancy facilities occupy 1.92%, cultivated land occupies 1%, and building land occupies 0.19%. The annual average runoff is 5.7 × 10^10^ m^3^, and the annual average precipitation is 536.1 mm [11]. At present, the upper reaches of the Jinsha River are planned to be a 13-level reservoir [10,11]. Due to the influence of water conservancy projects, the water area in the basin has increased by approximately 10%, and the construction land area has increased by approximately 227% since the 1990s [11]. According to the habitat characteristics, hydrological characteristics, and elevation changes of the river channels in the survey area, combined with the traffic situation in the survey area, a total of 21 sampling points were set up in this survey (Figure 1 and Table 1).

### 2.2. Sampling Method

As the high-altitude area was in the icebound period from November to April, the local meteorological data showed that the dry season usually occurred in June and the wet season in September in most parts of the basin. Therefore, field surveys were conducted at the sampling sites in September 2019 (wet season) and June 2020 (dry season).

Fish samples were collected with permission from the local fisheries department, and based on the different habitat characteristics of the river, different fishing methods were adopted in different habitats of the same sample point to ensure the diversity of the catch. In shallow water and fast-flowing water, single-layer gill nets (net length 20–50 m, net height 1–2 m, net mesh 1–6 cm) were mainly used for fishing. In deep water bodies and areas with slow water flow, a ground cage (0.3 m wide, 0.25 m high, 8 m long, 4 mm mesh) was used as the main fishing method, and a gill net was used as the auxiliary method. To reduce random sampling errors and make the catch data more comparable, the mesh specifications, netting method, and collection time used in each sampling were kept as consistent as possible, and three samples were taken at different sections of each sampling point. Gill nets and ground cages were placed at approximately 18:00 in the evening of the previous day and collected at approximately 6:00 in the morning of the next day. Thus, the sampling time was approximately 12 h.

All collected samples were stored in 10% formalin solution and brought back to the laboratory to identify the species, and the body length (accurate to 0.1 cm) and body weight (accurate to 0.1 g) were measured. Fish identification was based on ‘Fuana Sinica Osteichthyes Cypriniformes II’ [8] and ‘Fishes in the Jinsha Jiang River Basin, the upper reaches of the Yangtze River, China’ [11]. The endangered level of fish was determined by the ‘Red List of China’s Vertebrates’ [25]. Environmental surveys were conducted using a portable water quality analyser (YSI), rangefinder, and current metre at various points. Measurements included water temperature (T), dissolved oxygen (DO), pH, total dissolved solids (TDS), total phosphorus (TP), total nitrogen (TN), chlorophyll a (CHL), ammonia nitrogen (NH_3_-N), phosphate (${\mathrm{PO}}_{4}^{3-}$), and flow velocity (V). The average values of data measured in the wet and dry seasons are shown in Table 1.

### 2.3. Data Analysis

The Shannon index (*H*) and the Pielou evenness index (*J*) were used to evaluate the fish diversity at the collection sites. The composition of dominant species was evaluated by the index of relative importance (*IRI*) [26]. The calculation formula is as follows:*H* = −∑*p*_i_ log_2_
*p*_i_; *J* = *H*/log_2_*S*; *IRI* = (*N*% + *W*%) × *F*%
where *p*_i_ = *N*_i_/*T* is the proportion of the number of the ith fish in the sampling reach (*N*_i_) to the total number of fish tails in the sampling reach (*T*), *S* is the number of fish species in the sampling reach, *N*% is the percentage of the number of a certain fish in the total number of fish tails in the sampling reach, *W*% is the weight percentage of a certain fish in the total weight, and *F*% is the occurrence frequency. Fish with *IRI* ≥ 1000 were selected as the dominant species.

Nonmetric multidimensional scale analysis (NMDS) was used to analyse the spatial and temporal distribution characteristics of fish communities [27]. The species data were transformed by lg(*x* + 1) and then based on the Bray–Curtis similarity measure. According to the stress coefficient, the analysis results of NMDS can be measured. Stress < 0.05 indicates that the ranking is credible, 0.05 < stress < 0.10 indicates that the ranking is basically credible, 0.10 < stress < 0.20 indicates that the ranking results have some reference significance, and 0.20 < stress < 0.30 indicates that the sorting distribution is completely untrustworthy. Redundancy analysis (RDA) was used to rank fish biomass data and environmental factors [28,29], and a Monte Carlo substitution test was used to determine the importance and significance of environmental variables. To reduce the error caused by the large variance difference of the data, the fish biomass was Hellinger-transformed, and the environmental factors were normalized to make the data normally distributed [28,29].

Abundance biomass comparison curves (ABC curves) compare the quantity dominance curves with the biomass dominance curves in the same coordinate system and analyse the characteristics of communities under different disturbance conditions through the distribution of the two curves [30]. When the number dominance curve is below the biomass dominance curve, the community is in an undisturbed state, and the community is dominated by fish species with large individuals, slow growth, and late sexual maturation. When the number dominance curve crosses the biomass dominance curve, the community is in a moderately disturbed state, and the number of species with fast growth and small individuals increase. When the number dominance curve is above the biomass dominance curve, the community is in a state of serious disturbance, and the community is dominated by fast-growing species with small individuals [26,31].

The *W* statistic is used as a statistic of the ABC curve method [31], and its formula is as follows:$$W=\sum_{i}^{S}\frac{B_{i}-A_{i}}{50\left(S-1\right)}$$
where *A* is the cumulative percentage of biomass and quantity corresponding to the species number in the ABC curve, and *S* is the number of species occurring. When the biomass dominance curve is above the quantity dominance curve, *W* is positive; otherwise, *W* is negative.

The normalized biomass size spectrum (NBSS) uses the log_2_-converted upper limit of the particle size to divide the particle size as the abscess coordinate and the ratio of the biomass per unit area (m^2^) of the log_2_ converted to the width of the particle-size interval as the ordinate coordinate [32]. In the standardized type, the trend of points at each particle-size level is linear in the ideal state, and the slope is −1. When the community is disturbed, the spectral lines show a ‘dome’ parabola shape, and the curvature is affected by the productivity level, habitat environment, particle size, and fishing intensity [33]. If the size of the smallest fish species is V, then the first size class of the fish size spectrum is V~2V, the second size class is 2V~4V, and so on. The size intervals of the size classes increase in equal numbers, with 2 as the common ratio, and the fish are divided into different size classes according to their sizes [34,35]. Using the box diagram to judge whether there is a difference in the curvature of the trunk and tributaries, compare the box diagram of the trunk and tributaries. If the boxes do not overlap, the difference is very significant. If the median of both sides is outside the box range of the other side, the difference is significant. If only one median is within the range of the other party’s box or both medians are within the range of the other party’s box, the difference is not significant.

PRIMER 5.0 software (Shanghai, China) and one-way ANOVA in SPSS 19.0 software (Chicago, IL, USA) were used to test the significance of differences in the fish composition, ABC curve, and biomass particle-size spectrum in different seasons and branches. RStudio 3.5 software (Seattle, WA, USA) was used to analyse the correlations among the NMDS, RDA, environmental variables, and the curvature of the biomass particle-size spectrum.

## 3. Results

### 3.1. Composition of Fish

A total of 4062 fish, and 297,180.0 g, was collected in the two surveys. A total of 28 species was identified, and they belonged to 2 orders, 5 families, and 18 genera (Appendix A). There were 24 species of Cypriniformes and 4 species of Siluriformes. Among them, 14 species were Cyprinidae, 9 species were Cobitidae, 3 species were Sisoridae, 1 species was Balitoridae, and 1 species was Siluridae (Figure 2). *Schizothorax chongi*, *Gymnodiptychus pachycheilus,* and *Euchiloglanis davidi* have been classified as class II national protected species. ‘The Red List of China’s Vertebrates’ listed *Herzensteinia microcephalus*, *S. chongi*, *Gymnocypris potanini*, *E. davidi,* and *E. kishinouyei* as endangered fish and *S. dolichonema*, *S. kozlovi*, *S. prenantias*, *S. wangchiachii*, *Schizopygopsis malacanthus*, *Ptychobarbus kaznakovi*, and *G. pachycheilus* as vulnerable fish.

Ten new fish species were recorded in this survey in the upper reaches of the Jinsha River: Misgurnus anguillicaudatus, Paramisgurnus dabryanus, Pseudorasbora parva, Abbottina rivularis, Cyprinus carpio, Carassius auratus, Silurus meridionalis, Schizothorax prenanti, Schizothorax chongi, and Schizothorax davidi, and all were exotic fish. According to the habitat environment and migration mode of fish, fish in the upper reaches of the Jinsha River were divided into two ecological types: short migratory fish and settled fish. The dominant species type was settled fish, accounting for 64.29%.

### 3.2. Fish Diversity and Dominant Species

The Shannon index was 0.402–1.770, and the Pielou evenness index was 0.254–0.680 (Figure 3). According to the order of the *IRI* values, fish with *IRI* ≥ 1000 were the dominant species. The results showed that the dominant species of fish in the upper reaches of the Jinsha River were *T. stingura*, *S. wangchiachii*, and *S. malacanthus*, and there were no differences in their spatial and temporal distributions.

### 3.3. Temporal and Spatial Variation in Fish Communities

The Bray–Curtis similarity NMDS ranking method was used to analyse the fish community structure in the upper reaches of the Jinsha River. The results showed that the fish community structure in the upper reaches of the Jinsha River showed spatial autocorrelation, and the sites with similar geographical spaces were clustered into one group (Stress = 0.104, *p* < 0.05, Figure 4A). There were three groups: Group 1, S2–S5 and S7–S13 were clustered into a group, mainly with *P. kaznakovi*, *S. wangchiachii*, and *S. dolichonema*; Group 2, S14–S18 were clustered into another group, and the most common species were *S. wangchiachii*, *G. potanin*, *Jinshaia sinensi*, and *Pareuchiloglanis anteanalis*; and Group 3, S19–S21 were in the tributaries, and the main species were *S. wangchiachii*, *S. malacanthus*, *T. stenura*, and *Oreias dabryi*. The seasonal composition of the fish community at each sampling site changed significantly (Stress = 0.104, *p* < 0.05, Figure 4B). *G. pachycheilus*, *S. chongi,* and *E. davidi* were distributed in the wet season, and *O. dabryi*, *Triplophysa brevicauda*, *Triplophysa angeli,* and *Triplophysa yaopeizhii* were distributed in the dry season.

The RDA results in the dry season and wet season explained the relationship between the differences in fish community structure and environmental factors, but the main environmental factors affecting the changes in the fish community were different in different periods. In the wet season, the proportion of fish community variation explained by Axes 1 and 2 was 76.12%. The Monte Carlo displacement test showed that the fish community was affected by V and DO (*p* < 0.05, Figure 5A), in which *S. wangchiachii*, *P. kaznakovi*, *J. sinensis,* and *P. anteanalis* were mainly affected by V and *G. potanini*, *S. prenanti,* and *S. dolichonema* were mainly affected by DO. In the dry season, the proportion of fish community variation explained by Axes 1 and 2 was 77.66%. The Monte Carlo displacement test showed that the fish community was affected by ALT and DO (*p* < 0.05, Figure 5B), among which *P. kaznakovi* was mainly affected by ALT, while *G. potanini* and *S. kozlovi* were mainly affected by DO.

### 3.4. ABC Curve and Biomass Spectrum

The comparison curves of abundance/biomass showed that the W values were all less than 0 in the wet season and dry season. In the wet season, the abundance dominance curve was generally above the biomass dominance curve, and the starting point of the abundance dominance curve was higher than that of the biomass dominance curve, indicating that the fish community was seriously disturbed (Figure 6A). In the dry season, the biomass dominance curve intersected with the abundance dominance curve, indicating that the fish community was moderately disturbed (Figure 6B).

All regression results were significant (*p* < 0.05), and there were significant seasonal differences (Figure 6C,D) and spatial differences (Figure 7). In terms of seasonal variation, the curvature of the normalized biomass particle-size spectrum was −0.098 and −0.165 in the dry season and the wet season, respectively, and the curve was relatively gentle in the wet season (Figure 6C) and steep in the dry season (Figure 6D). Both curved domes were located in the particle-size range of 6–10, and the regression coefficients R^2^ were 0.77 and 0.95, respectively. In terms of the spatial variation, there were significant differences between tributaries (S19, S20, and S21) and main streams (S7, S8, S9, S13, and S14) in the normalized biomass grain size spectrum (*p* < 0.05, Figure 7). The curvature ranges of the tributaries were −0.1807 to −0.0398 and −0.1116 to −0.0016 in the dry season and wet season, respectively, and the R^2^ ranges were 0.91–0.99 and 0.93–0.99. The curvature ranges of the main stream in the dry season and wet season were −0.2407 to −0.1202 and −0.6241 to −0.1239, respectively, and the R^2^ ranges were 0.86–0.93 and 0.61–0.99, respectively (Table 2).

Correlation analysis showed that the curvature was positively correlated with the flow velocity (*p* < 0.01, R = 0.50) and DO (*p* < 0.01, R = 0.83) and was significantly negatively correlated with the EC (*p* < 0.01, R = −0.70) and TDS (*p* < 0.01, R = −0.70) (Figure 8).

## 4. Discussion

The fishing methods and fishing times were the same at each site in this study, and different fishing methods were arranged according to different habitats at the same site to ensure the diversity of fish sampled at each site. This study showed that the fish community structure in the upper reaches of the Jinsha River was generally consistent with the typical structure of the fish fauna of the Tibetan Plateau; namely, the three most abundant taxa were mainly Schizothoracinae, Sisoridae, and *Triplophysa* [6,7]. As far as the results of this study are concerned, native fishes dominated the catch in the upper Jinsha River, while exotic fish were mostly collected incidentally. This result indicates that the indigenous fish resources in the upper reaches of the Jinsha River are generally stable, and the invasions of nonnative fish are still in the early stages. In terms of ecological types, most fish in the upper reaches of the Jinsha River are omnivorous and benthic. This result may be due to the lack of prey resources and rapids in the plateau waters [36]. Compared with historical data [10,11], no native fish species, such as *Triplophysa leptosome*, *Schizothorax grahami*, *Lepturichthys fimbriata,* and *Pareuchiloglanis sinensis*, were collected in the two surveys, but the total species number increased by more than 10 species. The endemic fish species in the upper reaches of the Yangtze River increased from 9 to 12 species in the past, and the number of provincial protected fish species increased from 3 to 6 species in the past. The above reasons may be caused by religious release activities. In the upper reaches of the Jinsha River, where Tibetan Buddhism is prevalent, fish are one of the main animals released in religious activities. In recent years, with the development of society and the economy, an increasing number of fish have been sold in the market, and fish have gradually become one of the main objects of release [11]. Therefore, the 10-year ban on fishing in the Yangtze River should be accompanied by good supervision of fish release.

According to the general range of diversity index proposed by Magurran (1.5–3.5) [37], the Shannon index of fish in the upper reaches of the Jinsha River was 0.402–1.770, indicating that the fish diversity was deficient. The Pielou evenness index reflects the number and distribution of fish species in the water area. The lower the index is, the lower the complexity and stability of the community structure [36]. Therefore, the complexity and stability of the fish community structure in the upper reaches of the Jinsha River are relatively low. This low steady state is very fragile. Because most of these plateau fish have characteristics such as slow growth and late sexual maturity, once the population is excessively disturbed, it will be difficult to recover [10,11]. Fish communities generally have low species diversity and characteristics of high endemicity to local proportions in the upper reaches of the Jinsha River. The low water temperature contour elevation of river habitats in harsh environmental conditions limits the growth and reproduction of fish. Only a few species with strong adaptability to high-altitude habitats, such as Schizothoracinae, Sisoridae, and *Triplophysa*, can survive in these habitats [2,10]. The upper reaches of the Jinsha River are typical of a plateau river in the Hengduan Mountain area. The fish diversity and resources are low, which may be caused by the higher altitude.

The ABC curve showed that the abundance dominance curve was generally above the biomass dominance curve, indicating that the fish community structure in both seasons was in a state of interference, and the fish community composition was mainly dominated by fast-growing species with small individuals [26,31]. This is consistent with the survey that the dominant species of the whole year are mainly small fish, such as *Triplophysa* (sexual maturity at 3 months) and *S. malacanthus* (sexual maturity at 1 year), distributed in each river section of the basin [10,11]. In addition, the *W* value in the dry season was greater than that in the wet season, indicating that the fish community structure was more disturbed in the wet season. In the dry season, the temperature gradually rose after the ice period, but the temperature was still low, and human activities, such as the construction and operation of wading projects, were inhibited. However, the cold-water fish adapted to the plateau habitat gradually began to forage for bait in each river segment, and the community structure was relatively complete [38]. During the wet season, the temperature rises, and human activities are frequent. During this period, the tourist population increases substantially, the demand for wild fish increases continuously, and the fishing pressure increases sharply. Some areas, such as Batang County, have already seen overfishing phenomena [10,11]. The survey results show that there are sand dredgers of different scales in the confluence of tributaries such as Luoxu, Gangtuo, and Batang, which destroy fish habitats.

The spatiotemporal dynamics of fish composition and community structure were mainly caused by the heterogeneity of environmental factors at spatiotemporal scales. The RDA in this study showed that the main factors affecting fish distribution were the flow velocity, dissolved oxygen, and altitude. The results of this study showed that *H. microcephalus* was distributed only in the S1 sample site above 3000 m elevation, while *P. kaznakovi* was distributed in the sample site above 2000 m elevation, with the limitation of species elevation distribution. *Ptychobarbus* is a specialized class group, and *Herzensteinia* is a highly specialized class group. This phenomenon may be explained by the differentiation of nutritional and spatial niches in Schizothoracinae with different degrees of specialization [39]. However, the scientific explanation needs further analysis and demonstration. In addition, hydropower cascade development contributes to the main causes of species diversity and community structure changes. Since the implementation of the Jinsha River upstream cascade, the river flow rate dropped significantly, and the decrease in water reduced the river-wetted perimeter area, water flow velocity change, and channel depth; additionally, the dissolved oxygen and habitat changes resulted in reduced fish survival in the reservoir area [4,5]. In this study, we found that the temporal and spatial distributions of Schizothoracinae were positively correlated with flow velocity and dissolved oxygen, and there was more biomass and larger particle-size classes in tributaries with higher flow velocity and dissolved oxygen. The standardized biomass particle-size spectrum parameters of fish confirmed the above conclusion, which could reflect the ecosystem structure and function. When the fish community was in an undisturbed state, the curvature of the fish biomass particle-size spectrum was larger, and the curve was relatively flat. In contrast, when the fish community was disturbed, the curvature of the fish biomass particle-size spectrum was smaller, and the curve was relatively steep [13,14]. This study showed that the normalized biomass grain size spectrum of fish in tributaries was higher than that in fish in the main stream, indicating that the degree of disturbance of fish in tributaries was much lower than in the main stream. In addition, there was a significant correlation between the normalized biomass grain size spectrum and the environmental factors of the water area, especially for the flow rate and dissolved oxygen. The ecological environment of the water area can cause changes in the fish species composition structure at each grain level by affecting fish growth. The main stream has a high flow velocity and high dissolved oxygen. Some studies suggest that adult fish of the same species need higher flow velocity and dissolved oxygen to meet the physiological needs of spawning and baiting, which makes the adult fish biomass of tributaries greater and fish resources reach the maximum value of the sample point [40,41].

The life histories (such as reproduction) of most fish distributed in the upper reaches of the Jinsha River were fully or partially dependent on the flow-water habitat, such as short-migration Schizothoracinae fishes (*S. wangchiachii*, *S. dolichonema, S. kozlovi, and P. kaznakovi*) and current-loving cold water Sisoridae fishes (*E. kishinouyei*, *E. davidi*, *P. anteanalis,* and *P. sinensis*). Main stream cascade hydropower development has led to habitat barriers, habitat fragmentation, and habitat shrinkage in flowing water bodies. However, the resources of endemic fish in the upper reaches of the Yangtze River, such as *S. malacanthus*, *S. wangchiachii*, *S. kozlovi,* and *S. dolichonema*, are abundant in the tributaries of the upper reaches of the Jinsha River. Therefore, these tributaries have certain fish diversity protection value. A good habitat is fundamental to the maintenance and development of fish resources. In many basins where the main stream is strongly affected by hydropower development, tributaries with good natural connectivity can often serve as important habitats to maintain the diversity of fish in the main stream [42]. On the one hand, these tributaries can provide spawning and feeding grounds for the main stream fish and shelter for some rare fish [43]. On the other hand, after the reservoir is formed in the main stream, the flowing water habitat of the tributaries will become the habitat of rapid-stream fishes in the main stream [44] to alleviate the contradiction between hydropower development and water ecological protection in the Jinsha River basin. Currently, a tributary of habitat protection with more effective cases in the diversity of fish is rich in plain areas, and rich species diversity is the necessary premise of the implementation of a tributary of habitat protection [45]. However, because of the cascade development in the Jinsha River, habitat conservation strategies are necessary for tributaries with more fish diversity (restoring connectivity and expanding habitat protection for tributaries) [45].

The upper reaches of the Jinsha River are complex ecosystems that are mainly affected by human activities [6]. The degree of disturbance and the ecological characteristics of the fish community in the upper reaches of the Jinsha River were studied by combining the ABC curve and biomass particle-size spectrum. To some extent, the study of the fish biomass particle-size spectrum can increase the understanding of the changes in fish particle-size structure in the upper reaches of the Jinsha River and provide a new perspective to study the response of plateau aquatic ecosystems to natural disturbances and human activities. Because there has been little previous research on the Jinsha River upstream fish biomass particle-size spectrum, the comparability of historical data is limited, there is a lack of long-term historical studies of fish size structure, and there is not a good understanding or evaluation of the particle size of the fish community structure and the characteristics of the ecosystem evolution trend. Therefore, there is a need to further increase continuity surveys for long-term objectives to study the particle size of structure characteristics. The aim of this study was to better understand the status and ecological characteristics of fish communities in plateau water to promote the development and protection of fishery resources in the Yangtze River.

## 5. Conclusions

The fish diversity was at a deficient level, and flow velocity, altitude, and dissolved oxygen were the main influencing factors causing the differences in fish community structure in the upper Jinsha River. The fish communities were moderately or seriously disturbed. The degree of disturbance of fish in tributaries was much lower than that in the main stream. Compared with the historical data, the structure of the fish community in the Jinsha River changed significantly, with more exotic species and smaller individual fish. It is suggested that habitat conservation strategies (restoring connectivity and expanding habitat protection for tributaries) be adopted in the upper tributaries of the Jinsha River to provide a reference for the restoration of fishery resources and the conservation of fish diversity in the Yangtze River.

## Figures and Tables

**Figure 1 animals-12-03412-f001:**
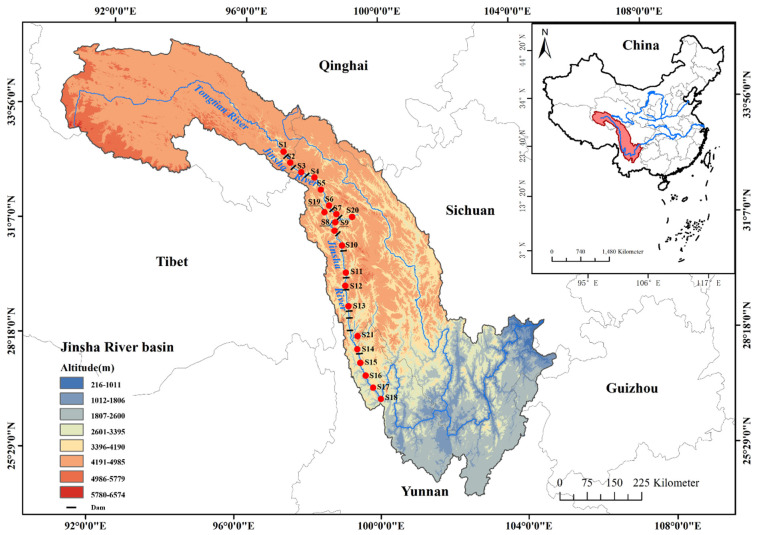
Map of sampling sites in the upper reaches of the Jinsha River.

**Figure 2 animals-12-03412-f002:**
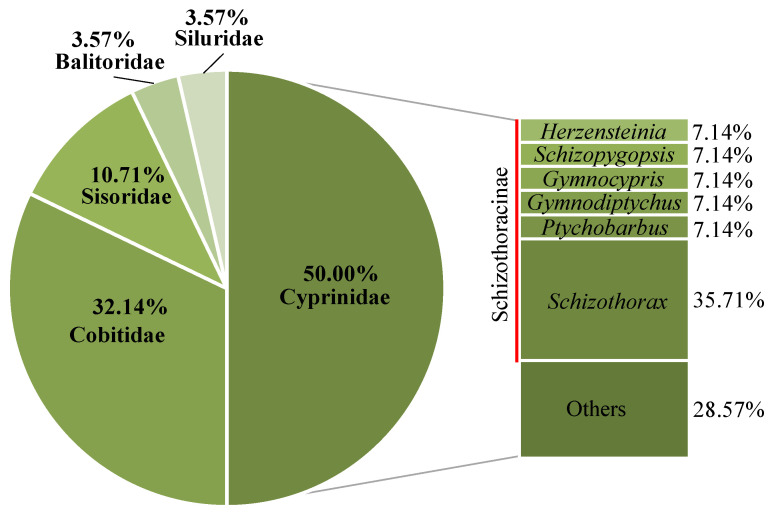
Proportion of fish families in the upper reaches of the Jinsha River.

**Figure 3 animals-12-03412-f003:**
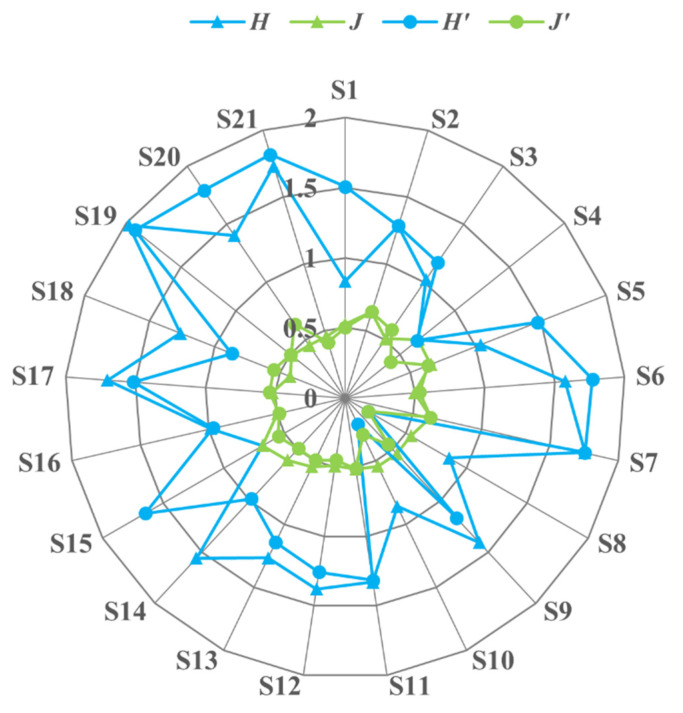
Fish biodiversity index of each sampling point. Note: The code information of the sampling points is shown in Table 1. Index of diversity in the wet season: *H*—Shannon index and *J*—Pielou evenness index; index of diversity in the dry season: *H′*—Shannon index and *J′*—Pielou evenness index.

**Figure 4 animals-12-03412-f004:**
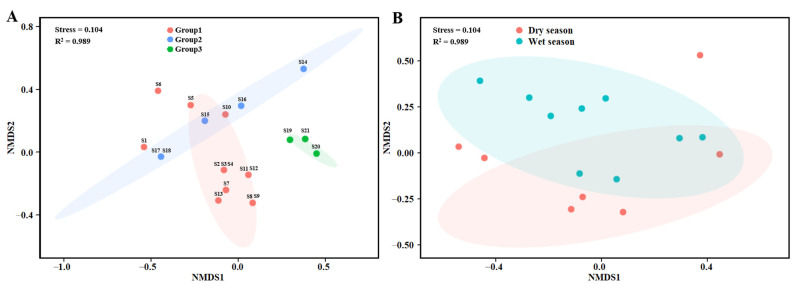
NMDS ordination of spatial (**A**) and temporal (**B**) variation in the fish community.

**Figure 5 animals-12-03412-f005:**
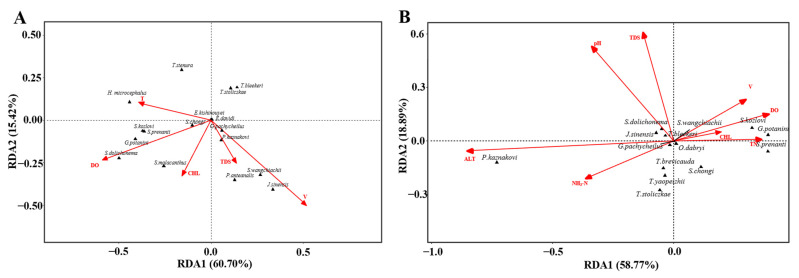
Redundancy analysis (RDA) of the relationship between fish distribution and environmental factors in the wet season (**A**) and dry season (**B**).

**Figure 6 animals-12-03412-f006:**
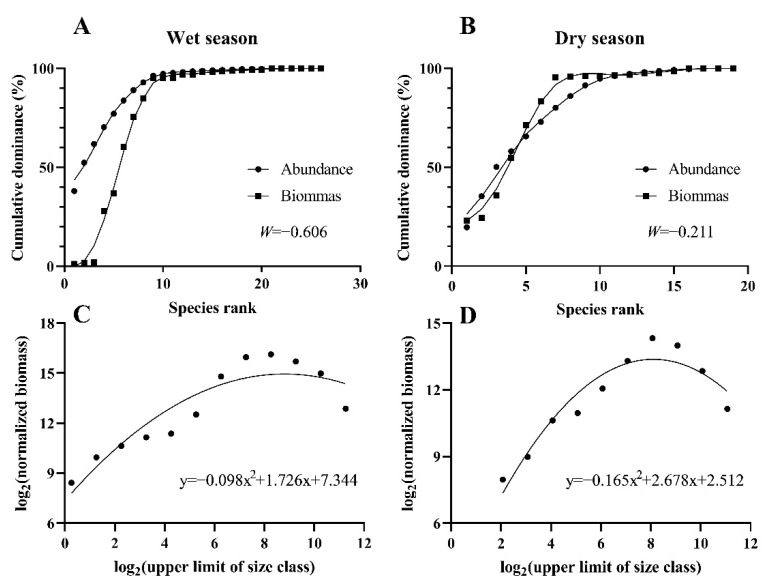
Abundance biomass comparison curves (**A**,**B**) and biomass spectrum (**C**,**D**) of the fish community in the upper reaches of the Jinsha River.

**Figure 7 animals-12-03412-f007:**
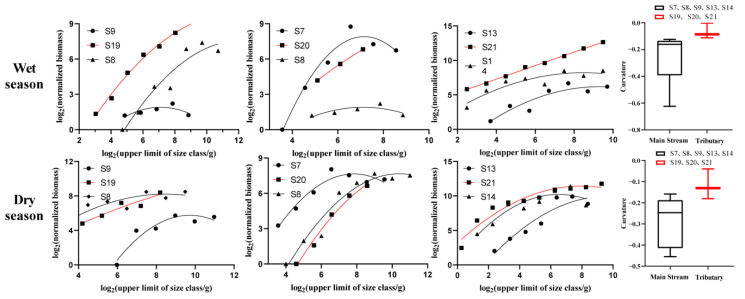
Standardized biomass particle-size spectrum of fish in the main stream and tributaries of the upper Jinsha River in different seasons and box plot of curvature of the main stream and tributary stream (horizontal line from top to bottom showing top edge, median, and bottom edge).

**Figure 8 animals-12-03412-f008:**
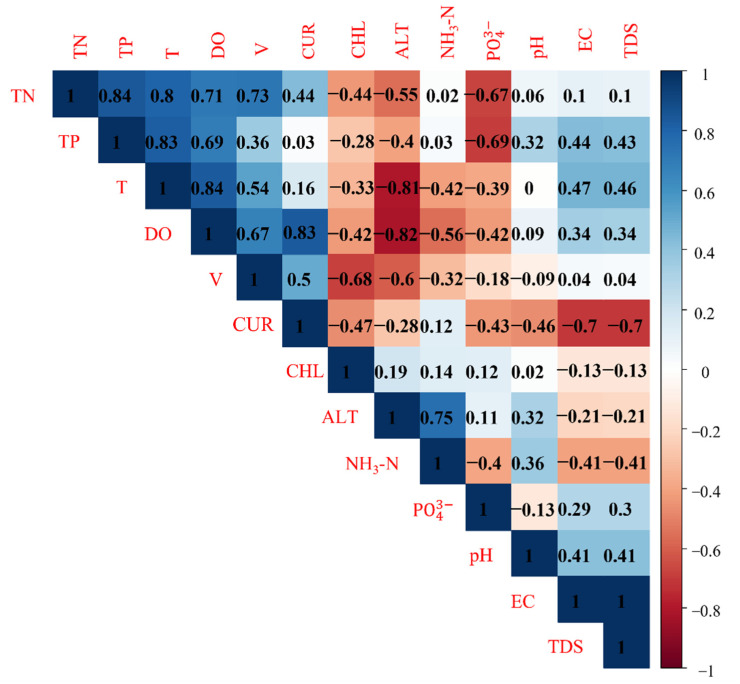
Correlation between environmental variables throughout the year and curvature of the upper reaches of the Jinsha River.

**Table 1 animals-12-03412-t001:** Spatial parameters and water quality of sampling sites in the upper reaches of the Jinsha River.

Site Code	Place Name	Province	Latitude and Longitude	Altitude (m)	T (℃)	DO (mg/L)	V (m/s)	pH	TDS (mg/L)	TP (mg/L)	TN (mg/L)	CHL (μg/L)	NH_3_-N (mg/L)	${\mathrm{PO}}_{4}^{3-}$ (mg/L)
S1	Zhimenda	Qinghai, Yushu	E97°14′53.67″ N33°0′22.88″	3535	11.85	7.27	1.65	8.56	832.00	0.2377	0.5685	0.27	0.2300	0.0034
S2	Benda	Sichuan, Shiqu	E97°30′50.28″ N32°39′4.50″	3424	12.00	7.46	2.10	8.53	744.25	0.1888	0.5251	0.25	0.1054	0.0058
S3	Luoxu	Sichuan, Shiqu	E97°59′43.47″ N32°27′41.20″	3288	13.30	7.17	1.55	8.38	656.50	0.3499	0.5532	0.35	0.1662	0.0049
S4	Maga	Sichuan, Dege	E98°8′27.40″ N32°22′47.58″	3253	13.00	7.26	1.65	8.41	708.50	0.2249	0.4916	0.40	0.1459	0.0047
S5	Kasongdu	Sichuan, Dege	E98°23′38.96″ N32°3′19.60″	3132	14.10	7.27	1.40	8.45	715.00	0.5377	0.5435	0.69	0.1968	0.0028
S6	Gangtuo	Sichuan, Dege	E98°34′18.51″ N31°38′41.61″	3044	14.25	7.08	1.95	8.35	728.00	0.5654	0.5358	0.46	0.1970	0.0032
S7	Zengqu	Sichuan, Baiyu	E98°53′3.44″ N31°22′48.22″	3027	13.80	7.00	0.80	8.36	581.75	0.3894	0.4586	0.58	0.1587	0.0058
S8	Jinsha	Sichuan, Baiyu	E98°46′18.16″ N31°15′8.94″	2982	13.35	7.19	1.40	8.48	457.60	0.2660	0.4490	0.34	0.1715	0.0067
S9	Boluo	Sichuan, Baiyu	E98°36′6.60″ N31°11′27.66″	2903	13.50	7.57	0.85	8.48	268.78	0.4198	0.5185	0.66	0.2031	0.0031
S10	Yebatan	Sichuan, Baiyu	E98°57′28.76″ N30°45′26.32″	2797	14.00	7.80	0.95	8.46	349.95	0.3523	0.5418	0.61	0.1744	0.0037
S11	Batang	Sichuan, Batang	E99°3′33.24″ N29°56′13.73″	2495	14.35	7.92	1.75	8.49	440.70	0.2832	0.5765	0.54	0.1261	0.0042
S12	Zubalong	Sichuan, Batang	E99° 0′38.54″ N29°46′17.46″	2476	15.50	7.86	1.30	8.40	529.75	0.5964	0.7221	0.29	0.1225	0.0038
S13	Suwalong	Sichuan, Batang	E99°3′57.06″ N29°25′42.45″	2394	15.85	8.38	1.65	8.39	507.00	0.5927	0.7541	0.14	0.1193	0.0032
S14	Benzilan	Yunnan, Deqin	E99°18′20.74″ N28°14′25.64″	2012	16.10	8.39	2.25	8.39	425.75	0.5755	0.9294	0.38	0.1253	0.0037
S15	Jiangdong	Yunnan, Deqin	E99°24′42.13″ N27°57′24.47″	1975	16.45	8.16	1.20	8.27	406.25	0.6247	0.7628	0.25	0.1227	0.0067
S16	Wujing	Yunnan, Deqin	E99°27′40.02″ N27°41′51.12″	1940	16.30	8.07	1.90	8.30	402.68	0.6035	0.8920	0.53	0.1933	0.0031
S17	Shangjiang	Yunnan, Deqin	E99°38′37.77″ N27°22′33.00″	1873	16.40	7.89	0.95	8.25	432.25	0.4494	0.6590	0.46	0.1423	0.0070
S18	Shigu	Yunnan, Yulong	E99°57′44.69″ N26°52′12.82″	1818	16.35	7.58	0.90	8.12	417.63	0.5136	0.8606	0.61	0.1772	0.0242
S19	Tongpu	Xizang, Jiangda	E98°23′23.86″ N31°35′27.64″	3242	13.85	7.01	1.30	8.40	146.91	0.4896	0.7534	0.37	0.3115	0.0025
S20	Hepo	Sichuan, Baiyu	E98°57′25.86″ N31°22′45.92″	3021	12.25	7.21	1.60	8.31	110.18	0.2086	0.4818	0.33	0.1692	0.0048
S21	Bendu	Sichuan, Derong	E99°18′1.84″ N28°36′51.49″	2293	14.30	7.38	1.20	8.24	130.98	0.2707	0.5226	0.52	0.1210	0.0050

Note: T, water temperature; DO, dissolved oxygen; V, flow velocity; TDS, total dissolved solids; TP, total phosphorus; TN, total nitrogen; CHL, chlorophyll a; NH_3_-N, ammonia nitrogen; ${\mathrm{PO}}_{4}^{3-}$, phosphate. The above water quality parameters are the average values in September 2019 and June 2020.

**Table 2 animals-12-03412-t002:** Spatial comparison of normalized biomass particle-size spectrum of fish in different seasons in the upper reaches of the Jinsha River.

Site	Wet Season			Dry Season			
	Fit Curve Equation	Curvature	R^2^	Fit Curve Equation	Curvature	R^2^	
S7	y = −0.6241x^2^ + 8.93x − 24.06	−0.6241	0.96	y = −0.2471x^2^ + 3.88x − 7.55	−0.2471	0.93	Main stream
S8	y = −0.1447x^2^ + 3.49x − 13.48	−0.1447	0.99	y = −0.1973x^2^ + 4.10x − 13.65	−0.1202	0.88	
S9	y = −0.1624x^2^ + 2.31x − 6.31	0.1624	0.61	y = −0.3741x^2^ + 7.27x − 29.59	−0.3741	0.90	
S13	y = −0.1608x^2^ + 2.99x − 7.68	−0.1608	0.85	y = −0.1586x^2^ + 3.05x − 4.76	−0.1586	0.90	
S14	y = −0.1239x^2^ + 2.10x − 0.67	−0.1239	0.86	y = −0.2160x^2^ + 2.81x + 1.05	−0.2160	0.86	
S19	y = −0.1116x^2^ + 2.65x − 5.82	−0.1116	0.93	y = −0.0398x^2^ + 1.33x − 0.04	−0.0398	0.91	Tributary
S20	y = −0.0857x^2^ + 2.37x − 5.67	−0.0857	0.99	y = −0.1807x^2^ + 4.13x − 15.34	−0.1807	0.99	
S21	y = −0.0016x^2^ + 1.00x + 3.30	−0.0016	0.99	y = −0.1307x^2^ + 2.08x + 3.17	−0.1307	0.93	

## Data Availability

Data will be available upon request.

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
