# Peer review of "Fish Community Structure and Biomass Particle-Size Spectrum in the Upper Reaches of the Jinsha River (China)"

_animals, 2022, doi:10.3390/ani12233412_

Round 1

Reviewer 1 Report

The concern for restoring fish resources and preserving fish diversity is meritorious.

The article is well written and the graphics are very suggestive.

I would like to ask the authors what exactly determined them to carry out this study? What changes did they notice in the analyzed habitat, prior to the research?

In line 210 it is stated that 7 new species were identified and in line 308, 10 species? Regardless of their number, how can this change in the structure of the fish community be explained?

Or in this experiment the fish were only identified, measured and weighed? Isn't that too little? However, it is about a number of 4062 slaughtered fish.

In Conclusions section, authors suggest the need to adopt some habitat conservation strategies in the upper tributaries of the Jinsha River, but they do not say anything about what exactly it could be?

Author Response

Thank you very much for your professional review of our articles. According to your comments, all the comments was revised. All modifications in the revised manuscript are marked in red color. All the response is shown below.

Reviewer 2 Report

The paper can be published after small corrections. Corrections are indicated as sticky notes on pdf. All corrections are recommended.

Author Response

On behalf of all the contributing authors, I would like to express our sincere appreciations of your letter and reviewers’ constructive comments concerning our article entitled “Fish community structure and biomass particle-size spectrum in the upper reaches of the Jinsha River (China)” (Manuscript No: animals-1981835). These comments are all valuable and helpful for improving our article. According to your comments, we have made extensive modifications to our manuscript. In response to your comments, we have made changes to our manuscript. In this revision, all changes to our manuscript are highlighted in red text in the document.

Reviewer 3 Report

Lines 53-54 – Please add some relevant international citation, e.g. from the Ecological Indicators journal

Lines 62-63 – The sentence is contradicting with references cited above in the Lines 60-61 (References #3-5) and below Line 74 (reference #11).

Line 120 – What month the investigation was conducted?  Authors wrote October 2020 (Line 30) and June 2020 (Line 120).

Lines 135-137 - Are there some differences with scientific names of the fishes given in the paper and by international databases like the Fishbase and the Eschmeyer’s Catalog of fishes?

Line 215 – “Sedentary” is not the same as “resident”. What did Authors mean?

Figure 3 – D and H indexes are strongly correlated each other. I recommend to provide some one of them to make the figure more comprehensive.

Figure 7 – What do the bars and horizontal lines show?

Lines 240-243 – What do the Authors mean writing “concentrations were found” and “distributions were found”?

General comment: It would be very informative if the Authors indicate what from the mentioned fish species were affected by direct human impact (fishery, recreational angling, hatchery and rearing).

Please, provide DOI for references

Author Response

(The authors gave the same response as above.)

Author Response

(The authors gave the same response as above.)

Round 2

Author Response

On behalf of all the contributing authors, I would like to express our sincere appreciations of your letter and reviewers’ constructive comments concerning our article entitled “Fish community structure and biomass particle-size spectrum in the upper reaches of the Jinsha River (China)” (Manuscript No: animals-1981835). These comments are all valuable and helpful for improving our article. According to your comments (Round 2), we have made modifications to our manuscript. In this revision, all changes to our manuscript are highlighted in red text in the document.
